# Imaging Markers in Genetic Forms of Parkinson’s Disease

**DOI:** 10.3390/brainsci13081212

**Published:** 2023-08-16

**Authors:** Amgad Droby, Avner Thaler, Anat Mirelman

**Affiliations:** 1Laboratory for Early Markers of Neurodegeneration (LEMON), Neurological Institute, Tel Aviv Medical Center, Tel Aviv 6801298, Israel; avnert@tlvmc.gov.il (A.T.); anatmi@tlvmc.gov.il (A.M.); 2Movement Disorders Unit, Neurological Institute, Tel Aviv Medical Center, Tel Aviv 6423906, Israel; 3Faculty of Medicine, Tel Aviv University, Tel Aviv 39040, Israel; 4Sagol School of Neuroscience, Tel Aviv University, Tel Aviv 39040, Israel

**Keywords:** imaging markers, genetics, Parkinson’s disease, prodromal PD

## Abstract

Parkinson’s disease (PD) is a complex neurodegenerative disorder characterized by motor symptoms such as bradykinesia, rigidity, and resting tremor. While the majority of PD cases are sporadic, approximately 15–20% of cases have a genetic component. Advances in neuroimaging techniques have provided valuable insights into the pathophysiology of PD, including the different genetic forms of the disease. This literature review aims to summarize the current state of knowledge regarding neuroimaging findings in genetic PD, focusing on the most prevalent known genetic forms: mutations in the *GBA1*, *LRRK2*, and *Parkin* genes. In this review, we will highlight the contributions of various neuroimaging modalities, including positron emission tomography (PET), single-photon emission computed tomography (SPECT), and magnetic resonance imaging (MRI), in elucidating the underlying pathophysiological mechanisms and potentially identifying candidate biomarkers for genetic forms of PD.

## 1. Background

Parkinson’s disease (PD) is a multifactorial disorder influenced by both genetic and environmental factors. PD is considered the second most common neurodegenerative disorder [1]. While the recorded incidence of PD may vary across the different studies, the prevalence of the disease in industrialized countries is estimated at 0.3% of the overall population, reaching 1% in individuals above 60 years of age [2]. The disease course is characterized mainly by a progressive loss of dopaminergic neuron innervation from the substantia nigra (SN) to the posterior basal ganglia, however, motor symptoms of the disease manifest clinically only when more than 50% of these affected nerves are degenerated [3]. Up to 15–20% of confirmed PD cases can be traced to identify genetic risk loci associated with PD that are prevalent in certain ethnic populations [4,5]. Hence, genetic forms of PD can offer a unique opportunity to explore specific underlying processes involved in disease pathogenesis.

Parkin (*PARK2*), PINK1 (*PARK6*), and DJ-1 (*PARK7*) gene mutations are frequent forms of autosomal recessive genes associated with the risk for PD. Clinically, patients harboring these mutations manifest with earlier age of disease onset and milder disease course with higher rates of responsiveness to levodopa treatment [4,6]. Homozygous mutations of glucocererosidase (*GBA1*) and leucine-rich repeat kinase 2 (*LRRK2*) are both prominent gene mutations that have been demonstrated to be strongly associated with PD risk. Mutations in the *GBA1* gene are identified as the strongest genetic risk factor for developing PD [7], accounting for 4–7% of cases. Clinically, *GBA*1-PD has been linked to a more severe disease phenotype with younger age of onset, increased neuropsychiatric symptoms, hallucinations, autonomic dysfunction, and cognitive decline [7,8,9]. To date, more than 50 mutations in the leucine-rich repeat kinase 2 (*LRRK2*) gene have been identified, however, G2019S was found to be the most prevalent mutation associated with PD with a frequency of 1% in sporadic and 4% in familial PD patients worldwide [10]), with higher incidence rates among Ashkenazi Jews and north African Berber Arabs (~20–40%) [11]. G2019S-PD patients demonstrate clinical characteristics that resemble sporadic patients, including the typical features of the first initial presenting motor symptoms of the disease and similar frequencies of non-motor features [12,13]. In these individuals, postural instability and gait disturbances seem to be more frequent than tremor as the presenting motor symptoms as well as the main motor syndrome [14,15,16]. Mutations in *PINK*, *GBA1*, and *LRRK2* have incomplete penetrance. Contrarily, *SNCA* (PARK1/PARK4) autosomal dominant gene mutations encode for α-synuclein; hence both point mutation carriers and triplication carriers show complete penetrance with earlier onset and more severe phenotype than duplication carriers [17]. Clinically, these patients demonstrate an early-onset disease with an aggressive disease course characterized by rapid motor progression and dementia.

Unlike Alzheimer’s disease (AD), where imaging markers for tracing amyloid-β up to 25 years prior to clinical symptom onset are available, to date, no synuclein tracer has been validated, thus other imaging modalities are used in PD. In vivo molecular neuroimaging methods including single-photon emission computed tomography (SPECT) and magnetic resonance imaging (MRI) are being increasingly used as supplementary diagnostic methods as well as for the investigation of underlying pathological processes [18]. However, their predictive value for early detection and progression monitoring of PD remains rather limited [1,19].

In this review, we summarize reported findings of recent imaging studies in genetic PD, focusing on *GBA1*-PD and *LRRK2*-PD, and assess the role of novel imaging-based methods as early markers for the disease.

## 2. Methods

We conducted a literature review using PubMed^®^ (https://pubmed.ncbi.nlm.nih.gov/ (accessed on June 2023)) using the search word strings including: [(MRI) AND (GBA)AND(parkinson’s disease)]; [(MRI) AND (LRRK2)AND(parkinson’s disease)]; [(nuclear imaging) AND (Genetics)AND(parkinson’s disease)]; [(nuclear imaging) AND (GBA)AND(parkinson’s disease)]; [(nuclear imaging) AND (LRRK2)AND(parkinson’s disease)]; [(nuclear imaging) AND (PARKIN)AND(parkinson’s disease)]; [((Neuroimaging) AND (Prodromal)) AND (Parkinson’s disease)]; [imaging biomarkers in prodromal PD]; [(DTI) AND (genetic Parkinson’s disease)]; [(DTI) AND (Prodromal Parkinson’s disease)]; [(DAT SPECT) AND (Parkinson’s disease genetics)]; [((MRI) AND (LRRK2)) AND (non-manifesting carriers)]; [((MRI) AND (GBA)) AND (non-manifesting carriers)]; [(FDG-PET) AND (genetic Parkinson’s disease)]; [(FDG-PET) AND (genetic Parkinson’s disease)]; [(FDG-PET) AND (Prodromal Parkinson’s disease)]; [((DAT SPECT) AND (GBA)) AND (Parkinson’s disease)]; [((DAT SPECT) AND (LRRK2)) AND (Parkinson’s disease)]; [((DAT SPECT) AND (PARKIN)) AND (Parkinson’s disease)]; [((FDG-PET) AND (PARKIN)) AND (Parkinson’s disease)]; [((FDG-PET) AND (PARKIN)) AND (Parkinson’s disease)]; [((FDG-PET) AND (LRRK2)) AND (Parkinson’s disease)]; [((FDG-PET) AND (GBA)) AND (Parkinson’s disease)]; [(functional imaging) AND (GBA Parkinson’s disease)]; [(functional MRI) AND (LRRK2) AND (Parkinson’s disease)]; [(functional MRI) AND (GBA) AND (Parkinson’s disease)]; [(neuromelanin MRI) AND (Parkinson’s disease)]; [(neuromelanin) AND (GBA) AND (Parkinson’s disease)]; [(neuromelanin MRI) AND (LRRK2) AND (Parkinson’s disease)]; [(neuromelanin MRI) AND (LRRK2) AND (Parkinson’s disease)]; [‘neuroimaging biomarkers in prodromal PD’]. The search yielded 1049 results in total. These were further filtered based on date of publications (articles published from 2000–2023 were considered), duplicates, case studies, being written in English, whether reporting findings in overall PD, and other conditions and/or diseases, yielding 94 articles on which this literature review is based on.

## 3. Imaging Studies Findings in Genetic Forms and Prodromal Phase of Parkinson’s Disease 

### 3.1. Genetic Forms of Parkinson’s Disease

#### 3.1.1. Molecular Imaging (SPECT and PET)

Positron emission tomography or single-positron emission tomography (SPECT)-based molecular imaging approaches are used to depict nigrostriatal dopaminergic neuronal dysfunction [20]. The imaging methods differ in the type of tracer used for imaging. The source of the signal in PET images originates from the decay of the used radiotracers, which produce positrons. These positrons react with electrons (annihilation) in the tissue, producing a small amount of energy in the form of two photons that travel in opposite directions, which are picked up by the detectors in the PET scanner [21,22]. In SPECT, gamma rays are emitted from the injected radiotracer molecules. The camera detectors mounted on a rotating gantry, allowing the detectors to be moved around the participant and/or imaged body part [23], can directly detect these gamma rays. 

A retrospective study analyzing (123I-FP-CIT) dopamine transporter (DAT) from monogenic PD patients with mutations in either *GBA1*, *LRRK2*, *SNCA*, *PINK1*, or *Parkin* reported greater asymmetrical striatal dopaminergic uptake in *GBA1* and *LRRK2* carriers compared with SNCA, *PINK1*, and *Parkin* carriers [24]. A recent dual-phase [^18^F]-FP-CIT PET (DAT) study by Kim et al. examined striatal dopamine and cerebral perfusion in *GBA*1-PD compared to sporadic PD (sPD). Decreased perfusion in cerebral subregions including the parietal and occipital regions in *GBA1*-PD compared to sPD was detected. These differences coincided with higher frequencies of non-motor symptoms including REM sleep behavior disorder (RBD) and psychiatric symptoms in *GBA1*-PD [25]. Interestingly, unlike the reported findings in [24], this study demonstrated no significant differences in the measured striatal DAT uptake were detected between *GBA1*-PD and sPD [25]. These discrepancies might be attributed to different mutation types assessed in each study or to the variability of PD severity [24,25]. 

Consistently, in a recent study we also found similar striatal DAT binding ratios (SBRs) among N = 15 *LRRK2*-PD, N = 16 *GBA1*-PD, and N = 15 sPD early-stage patients with a mean disease duration of 2.6 years [26]. However, in a multisite cross-sectional study by the Parkinson’s Progression Markers Initiative (PPMI) study group consisting of N = 158 *LRRK2*-PD (89% G2019S), N = 80 *GBA1*-PD (89% N370S), and N = 361 sPD early-stage patients (mean disease duration of 2.9 ± 1.9 years; 3.1 ± 2 years; and 2.6 ± 0.6 years, respectively), more (better) SBR DAT binding was observed in both genetic PD groups compared to sPD, restricted contra-laterally to the more affected side of the body [27], despite similar motor and cognitive symptoms between the groups. The authors speculated that this finding could be attributed to a slower rate of DAT decline in genetic variants of PD, or this might be a result of dopamine release disruption prior to dopamine terminals loss [27]. Nevertheless, the biological basis underlying this observation remains to be determined. 

In a retrospective analysis, the temporal trajectories of measured putaminal SBR values were modeled over a time period spanning 10 years during the premotor phase of PD from N = 367 sPDs, N = 72 *LRRK2*-PDs (G2019S), and N = 39 *GBA1*-PDs (N370S) from the PPMI registry. *GBA*1-PD patients demonstrated a more rapid deterioration as reflected by the rate of annual change in the measured putaminal SBRs throughout the prodromal phase of the disease. Based on these results, the authors argued that the onset of striatal DAT progression could vary depending on the individual genetic background, possibly reflecting the effects of distinct compensatory mechanisms in these individuals [28]. 

[^18^F]-fluorodeoxyglucose (FDG)-PET studies in genetic PD patients have documented widespread metabolic alterations in *GBA1*-PD patients with N370S/R496H mutations encompassing frontal, parietal, striatal, and thalamic brain regions [29]. Schindlbeck et al. applied graph theory methods to investigate the genotype effects on the organization of PD-related networks in N = 12 *GBA1*-PDs, N = 14 *LRRK2*-PDs, N = 14 sporadic PDs (sPDs), and N = 14 healthy controls using FDG-PET. While all three PD groups showed similar disease networks, the location and modular distribution of the network connections differed across groups. For example, *LRRK2*-PDs gained connections within the network core, with the formation of distinct functional pathways between the cerebellum and putamen. In *GBA1*-PD, the majority of functional connections were formed outside the core of the network, involving cortico-cortical pathways at the network periphery. The authors concluded that these distinct connectivity patterns might give rise to the less aggressive disease course in *LRRK2*-PDs compared to that observed in *GBA*1-PDs [30].

#### 3.1.2. MRI Findings

*Volumetric measures*: A study comparing N = 9 *Parkin*-PDs with N = 14 sPDs detected increased gray matter (GM) volume in the right globus pallidus externus, the head of the left caudate, and right putamen in *Parkin*-PD compared with sPD [31]. A different study investigated patterns of GM alterations in N = 9 *Parkin*-PD patients and reported reduced bilateral caudate volumes compared to age-matched sPD individuals. These changes were not found to be associated with clinical or behavioral differences between the study groups [32].

Voxel-based morphometry (VBM) was used to examine structural changes in N = 10 PD patients with mutations in the *LRRK2* gene, N = 10 *LRRK2*-non-manifesting carriers (NMC), N = 24 sPD, and N = 12 non-manifesting non-carrier (NMNC) healthy individuals. Higher GM volumes in the cerebellum and left precentral gyrus was observed in the *LRRK2*-PD group compared to age- and-sex matched sPD and healthy controls, as well as a significant GM volume decrease in bilateral putaminal volumes when comparing *LRRK2*-PDs vs. age-matched healthy controls. The authors argued that these observed patterns of GM volume alterations might be indicative of prominent compensatory mechanisms within motor networks of the brain in *LRRK2*-PD, related to the more benign disease course [33]. Our group examined cortical thickness parameters between three groups of PD patients (sPDs, *LRRK2*-PDs, and *GBA*1-PDs) compared to three groups of unaffected healthy participants (NMNCs, *LRRK2*-NMCs, and *GBA*1-NMCs). No structural differences were observed among the different groups of NMCs, and no differences in cortical thickness and deep gray matter (DGM) volumes were observed between the genetic PD groups and sPD [34]. However, there was significant reduction in cortical thickness between all PD participants compared with the non-PD participants, indicating that widespread cortical and subcortical GM atrophy occurs in PD, but this is less likely in the premotor stages of the disease or in individuals with genetic mutations associated with PD risk. Another study investigated cortical thickness differences among different early-stage GBA1-PD variants including N370S, E326K, and T369M compared to age-matched sPD and HCs from the PPMI database. The obtained results from this study showed a significant post-central gyrus cortical thickness reduction in GBA1 (N370S)-PD group compared to GBA1 (E326K & T396M)-PD. Based on these findings, the authors argued that distinct patterns of degeneration rates might be observed in early-stage GBA1-PD variants [35]. A recent study investigated longitudinal patterns of cortical thickness over a time period of 5 years in N=10 GBA1 (D409H)-PD and N=20 sPD patients. The authors reported demonstrated a widespread bilateral cortical thickness reduction involving the bilateral supramarginal and superior frontal gyri, left precentral, and middle frontal and temporal regions, as well as the precuneus and parietal gyri which coincided with greater disease severity progression as reflected by Hohn & Yahr and the unified Parkinson’s disease scale (UPDRS) total and sub-scores [36]. 

*Neuromelanin-sensitive (NM) MRI:* NM-MRI enables the assessment of the extent of dopaminergic neuronal damage in the SN. In these neurons, neuromelanin plays a protective role by chelating excess iron [37]. Changes in NM-SN patterns have been examined in several studies. For example, NM was found to be lower in PD patients compared to age-matched healthy controls, coinciding with motor asymmetry and disease severity [38,39,40]. Interestingly, varying results were obtained when examining NM in different genetic forms of PD. A study investigating NM-SN between sPD and *PARK2*-PD patients reported reduced NM-SN in the *PARK2*-PD group compared to sPDs, however, no significant correlations between the measured NM-SN signal and clinical motor scores were reported [41]. In a recent study including *LRRK2*-PD and sPD participants compared to healthy controls, the authors investigated NM within brainstem structures including the SN, locus coerluleus (LC), and red nucleus (RN). These authors reported NM loss in the ventrolateral tier of the SN of both PD groups. Furthermore, while the measured NM-SN was a good discriminator between PD patients and controls, the NM-LC was found to best discriminate among PD subgroups with *LRRK2*-PD showing preserved NM-LC content compared to sPD. Based on these observations, the authors speculated that NM-based markers can serve as candidate markers for phenotypic PD features [42]. In a recent study, we investigated NM-SN differences among sPD, *LRRK2*-PD, and *GBA1*-PD patients. The findings indicated genotype-specific differences in NM-based radiomic features only. Specifically, significant differences in the left and NM-SN skewness radiomic feature between sPD were observed, possibly indicating greater dopaminergic loss in the *LRRK2*-PD group [26].

*Diffusion tensor imaging (DTI):* DTI has become one of the most popular MRI techniques in brain research, as well as in clinical practice, and is being constantly validated and developed in terms of acquisition schemes, image processing, analysis, and interpretation [43]. DTI provides quantification of water-molecule diffusion to reflect the integrity of the white matter structure and enables visualization and characterization of white matter (WM) fasciculi in 3D [44]. The method measures diffusion in multiple directions and, using tensor decomposition, extracts the diffusivities parallel (λ//) and perpendicular (λ⊥) to the fiber, which are used to calculate DTI scalars such as fractional anisotropy (FA), mean diffusivity (MD), radial diffusivity (RD), and axial diffusivity (AD).

In an earlier study, WM microstructural differences were investigated in N = 15 *GBA*1-PD an N = 16 sPD patients. The authors identified reduced FA in the inter-hemispheric and intra-hemispheric bundles including the corpus callosum, olfactory tract, cingulum, and internal and external capsules in the *GBA1*-PD group. The measured mean FA along the corpus callosum, external capsule, and the olfactory tract was found to be associated with verbal fluency in the overall patient group. No significant correlations between further clinical and DTI scores were obtained in the overall group as well as in the *GAB1*-PD subgroup [45]. These regions were previously reported to be affected in patients with PD compared with healthy controls in a meta-analysis based on 39 imaging studies [46]. A recent study by Yu et al. investigated WM alteration in *PARKIN-*PD using tract-based spatial statistics (TBSS) in N = 12 *PARKIN*-PDs, N = 14 sPDs, N= 15 *PARKIN*-NMCs, and N = 13 NMNCs. Compared to sPD, *PARKIN*-PD demonstrated significantly lower FA in the cortico-spinal tract, bi-lateral external capsules, genu of the corpus callosum, and the middle cerebellar peduncle and higher MD values in the left interior limb of internal capsule, right cortico-spinal tract, and the left externa capsule. Interestingly, no significant WM differences were detected between *PARKIN*-NMCs and NMNCs [47]. While no significant interactions between genotype and disease status were detected, the authors concluded that *PARKIN*-PD manifests more severe WM damage encompassing fibers implicated in motor and cognitive functioning [47].

*Functional MRI (fMRI):* fMRI is a non-invasive imaging technique enabling the assessment of brain activity. In many neurological diseases, including PD, fMRI experiments have contributed greatly to the understanding of functional plasticity mechanisms taking place parallel to the neurodegenerative process [34,48]. Brain activation patterns in N = 9 *PARKIN*-PD and N = 11 sPD patients while “ON” medication were investigated using a simple motor sequential finger task. The researchers of this study reported no between-group differences in task performance or cerebral activation patterns, possibly indicating equal response to dopaminergic therapy in the study groups [49].

*Resting-state fMRI:* Resting-state fMRI (rs-fMRI) studies mainly aim to investigate functional connectivity (FC) patterns within major functional networks of the brain [50,51]. Rs-fMRI study paradigms are task-negative and require no active participation of the subjects assessed, avoiding confounders such as the tasks’ demands, rendering this method appealing. This method is widely applied for the investigation of many neurological and neuropsychiatric conditions including PD [34,48].

Using rs-fMRI, FC levels within major brain functional networks were examined in N= 8 *PINK1* and *PARKIN*-PD patients compared to N = 12 NMC and N = 22 NMNCs. The authors reported lower FC among both PD and NMC compared to NMNCs. PD patients demonstrated lower FC compared with NMCs in the right fronto-parietal network and the executive network, which was associated with their worse cognitive performance [52].

FC within striatal regions was investigated in N = 11 drug-naïve *LRRK2*-PD patients (R1628P and G2385R mutations carriers) compared to N = 11 drug-naïve sPD patients and N = 22 age-matched healthy controls using a seed-based approach. *LRRK2*-PD demonstrated reduced FC between the left anterior putamen and the right calcarine gyrus, the right anterior putamen and bilateral calcarine gyri, left posterior putamen and bilateral superior frontal gyri, and right posterior putamen and bilateral precuneus when compared with the sPD group. Moreover, *LRRK2*-PD showed lower FC levels between the left posterior putamen and the sensorimotor cortices compared to healthy controls [53]. The authors concluded that these FC alterations observed within the cortico-striatal circuits in the *LRRK2*-PD group could be a distinct clinical phenotype in these patients. It is worthwhile to point out that the investigated patients in this study were of a relatively younger age, averaging less than 50 years, and had short disease duration.

A multimodal imaging study explored imaging differences in PD patients with different *GBA*1 variants. The findings indicated reduced F-DOPA uptake in the bilateral caudate nuclei, the ipsilateral antero-medial putamen, and the contra-lateral nucleus accumbens relative to the affected side in *GBA1*-PD (p.E326K and p.T408M) carriers compared to sPD. Meanwhile, only minor clinical differences were observed between *GBA1*-PD and sPD. On FDG-PET, *GBA1*-PD showed significant reduction in the bilateral medial and lateral parietal lobe compared to sPD. On rs-fMRI, reduced FC was observed between left and right caudate and the bilateral occipital cortex and between the right nucleus accumbens and the left superior parietal cortex and right fusiform gyrus in *GBA1*-PD vs. sPD [54]. The authors concluded that the imaging findings could provide a confirmation of the more severe disease course in *GBA1*-PD patients and their higher susceptibility for executive function decline, dementia, and visual hallucinations [54]. Nonetheless, this study presents findings in only two mutation variants associated with the *GBA1* gene.

See Figure 1 for a graphic illustration of key findings from imaging studies in *GBA1*- and *LRRK2*-PD patients.

### 3.2. Imaging Findings in Non-Manifesting Carriers (NMCs) of PD-Related Genetic Mutations

Key findings from nuclear imaging and MRI studies in *GBA1*- and *LRRK2*-NMCs are summarized in Figure 1.

#### 3.2.1. Molecular Imaging (SPECT and PET)

Combining FDG-PET and ^11^C-2β-carbomethoxy-3β-(4-fluorophenyl) tropane (CFT), cerebral metabolism and dopaminergic neuronal activity were assessed in N = 6 *GBA1* mutation carriers with and without Parkinsonism [55]. *GBA1*-PD showed significant reduction in cerebral glucose metabolism encompassing the medial surface of the frontal cortex and extending to the supplemental motor area (SMA) and cerebral hypo metabolism in the parieto-occipital cortices. Asymptomatic *GBA1* carriers (N = 3) demonstrated increased dopaminergic activity in the caudate nucleus based on ^11^C-CFT imaging. While no associations between the measured FDG-PET binding and clinical measures in the asymptomatic carriers were reported, the authors argued that observed cerebral hypo-metabolism patterns might underlie certain motor symptoms such as akinesia and the increased dopaminergic activity in the caudate might reflect presynaptic dopamine transporter upregulation or increased endogenous dopamine secretion in *GBA1* carriers [55].

Comparable striatal ^18^F-flurodopa (FDOPA) binding and reduced DAT-SBRs were observed in N = 25 *LRRK2*-NMCs compared to N = 35 healthy controls; however, these observed levels in the NMC groups were significantly higher than those of *LRRK2*-PD [56] The authors of this study speculated that this finding might be representative of a primary effect on DAT function where preserved FDOPA can persist up to symptom onset [50]. Similar findings were reported in a DAT-SPECT study, see [57], which reported evidence of striatal dopaminergic neuronal terminal loss in presymptomatic (G2019S) *LRRK2*-NMCs compared to NMNCs, indicating that this group might reflect an endophenotype where several of these NMCs would convert to definite PD [56,57]. Recent findings based on data from a PPMI study reported higher striatal SBRs in *GBA1-*NMCs compared with *LRRK2*-NMCs and healthy controls [27]. The direct effects of genetic variants on measured SBR levels remain not fully understood, and the obtained findings might suggest various patterns of pathological and compensatory mechanisms that simultaneously take place in the different genetic groups affecting DAT-SBRs. Therefore, understanding DAT downregulation and functional compensation dynamics in the preclinical phases of the disease is of great interest [27,58].

#### 3.2.2. MRI Findings

Subtle structural alterations in non-manifesting carriers of PD-related mutations were explored in many studies [31,59,60,61]. Morphometric fingerprints were assessed in *Parkin*-NMCs and *PINK1*-NMCs compared to age- and-sex matched NMNCs. The authors reported bilateral GM increase in the posterior putamen and the internal globus pallidum among NMCs irrespective of genotype. The authors concluded that this observed striatal volumetric increase might be related to excessive neuronal activity or reflect an adaption mechanism to compensate for dopaminergic activity in these individuals [59]. A study assessing seven *SNCA-*NMCs before and after development of PD compared the participants with N = 10 healthy controls. At baseline, *SNCA*-NMCs did not differ significantly from controls in caudate volume. However, at follow-up, *SNCA*-PDs demonstrated significantly lower caudate volumes compared to controls. Conversion to definite PD was not associated with cortical atrophy [62]. In a different study comparing N = 18 carriers of the rs356219 risk variant in the *SNCA* gene to N = 13 non-carriers, no significant differences in measured cortical thickness were detected between the investigated genetic groups [63].

In another study, brain patterns of volumetric brain alterations in N = 14 non-manifesting family members, all carriers of *PINK1* demonstrating psychiatric symptoms, compared to age- and gender-matched healthy individuals were investigated [60]. The authors of this study reported decreased GM volumes in the right hippocampus and para-hippocampal gyrus. In these individuals, the duration of psychiatric symptoms was further found to be associated with widespread frontal and limbic GM volume decrease [60]. In a combined VBM and TBSS-DTI study, volumetric and microstructural WM changes were investigated in N = 25 *LRRK2*-NMCs and N = 30 NMNCs. No significant GM differences between the study groups were detected. In DTI, *LRRK2*-NMC showed higher FA and lower AD and MD in the substantia nigra, as well as lower AD and MD in the nucleus accumbens. However, these observed differences did not survive correction for multiple comparisons. Despite the small effect size of the study, the authors argued that these findings might be indicative of subtle structural reorganization in *LRRK2*-NMC, possibly reflecting structural compensation [61].

The microstructural changes responsible for the observed patterns of GM volume increases in mutation carriers with PD remain not fully clear [64]. Such GM increases might reflect ongoing compensation to counteract the ongoing neurodegenerative process in the early stages of the disease. Alternatively, the obtained results from the various studies so far might be indicative of a non-linear change in cortical and subcortical volume/thickness depending on disease stage, where hypertrophy precedes the process and volumetric changes follow suit. The latter might render between-group comparisons less sensitive to detect subtle brain changes due to the relatively high within-group variance. In addition, small sample sizes, different acquisition and processing protocols for MRI datasets, and the lack of significant behavioral correlations impose further limitations in interpreting and generalizing these findings [34].

*NM*-*MRI*: To our knowledge, only one study has examined SN-NM in asymptomatic *LRRK2* and *GBA1* carriers. In this study, the authors compared the measured NM within the SN in N = 34 NMNCs, N = 24 LRRK2-NMCs, and N = 23 GBA1-NMCs. The obtained results indicated no significant differences between these study groups. Moreover, this study reported reduced NM in PDs compared to asymptomatic individuals, further highlighting that NM-based MRI markers can serve as sensitive tools for identifying the symptomatic phase of the disease [26,38,40].

*fMRI*: Functional imaging tools are useful for elucidating functional compensation coinciding with subtle dopaminergic decline and the appearance of non-motor symptoms in the prodromal phases of the disease or in individuals “at risk”. The occurrence of such functional changes in the prodromal phase was documented in numerous fMRI studies [65,66,67,68,69,70,71,72,73,74,75,76,77,78,79]. For example, a study including N = 37 G2019S-*LRRK2* NMCs and N = 32 NMNCs explored corticospinal connectivity patterns using rs-fMRI. The study detected connectivity changes among the *LRRK2*-NMCs with reduced functional connectivity between the right inferior parietal cortex and the dorsoposterior putamen but increased connectivity with the ventroanterior putamen, as compared with non-carriers. This shift in functional connectivity increased with age in *LRRK2*-NMC [71]. Similar findings were reported in an rs-fMRI study comparing FC profiles of the posterior putamen in N = 41 sPD patients compared to N = 36 matched healthy controls [69]. A graph-theory rs-fMRI study assessed network integrity in major brain functional networks such as the default mode network (DMN), salience network, dorsal attention network (DAN), and motor network in G2019G *LRRK2*-NMCs and NMNCs. *LRRK2*-NMCs demonstrated decreased network organization compared with NMCs, however, no differences between both groups were observed within the motor network [77]. Together, these findings are indicative of a compensational process within the motor network as a result of basal ganglia processing breakdown, possibly due to reduced dopaminergic input from the midbrain, in order to maintain performance due to dopaminergic depletion [34,69,73,77]. In a recent multimodal study combining DAT-SPECT and rs-fMRI, we investigated the association between presynaptic dopamine activity and striatal FC patterns in *GBA1*-NMCs and *LRRK2*-NMCs compared to NMNCs. The findings of the study indicated distinct group-specific patterns, in which *LRRK2*-NMCs showed significantly reduced SBRs in the right putamen compared to NMNCs, while no difference in measured SBRs was observed between *GBA1*-NMCs and NMNCs. In rs-fMRI, the opposite pattern was observed; namely, *LRRK2*-NMCs showed higher intra-striatal FC within the putamen compared to *GBA1*-NMCs. Furthermore, these measured DAT-SPECT SBRs were found to inversely correlate with the MDS research criteria for prodromal disease likelihood ratio (LR) scores [80,81] in the *GBA1*-NMC group; whereas rs-fMRI FC levels were associated positively with LR scores in the *LRRK2*-NMC group. Based on these findings, we concluded that differential DAT-FC patterns could be observed among the different genotypic groups in the prodromal phase of the disease, paving the way for future longitudinal investigations to further examine the potential role of FC-based measures for risk prediction in the early stages [79].

Numerous task-fMRI studies were carried out on NMCs of PD-related mutations aimed at exploring specific functional and cognitive aspects associated with the genetic status of risk for PD. For instance, *Parkin*-NMCs and *PARK1*-NMCs were found to recruit additional motor regions compared with NMNCs while performing a motor task, coinciding with comparable task performance. This might indicate that the recruitment of cortical motor areas might have a compensatory role in maintaining normal task performance indicative of early reorganization processes taking place in the “at risk” group [67]. This notion was further supported by a motor imagery study performed on *LRRK2*-NMCs. While error rates and reaction times did not differ between the groups, *LRRK2*-NMCs showed reduced motor imagery-related activity in the right caudate, indicating functional impairment in the striatum and increased activity in the right dorsal premotor cortex, which was counteracted by increased FC between this region and the right extra-striate body [70]. A study utilizing two different cognitive tasks reported that while there were no differences in reaction times, *LRRK2*-NMCs showed recruitment of three additional right-sided brain structures, inferior parietal lobe, precuneus, and the fusiform gyrus, compared to NMNCs while performing an attention interference (Stroop) task. The authors proposed that this observed right lateralized activation increase might be reflective of ventral attention network recruitment in these individuals [72,78]. Another study investigated attention and working memory performance using both Stroop and N-back tasks in *GBA1*-NMCs compared to *LRRK2*-NMCs and NMNCs. In line with previous findings, no performance differences were observed between the three study groups. Nevertheless, right medial frontal gyrus activation was observed in the *GBA1*-NMC group, where this region showed a task-negative activation pattern in both *LRRK2*-NMC and NMNC groups [75].

## 4. Discussion

PD is among the leading neurological diseases with highly effective symptomatic therapies available, yet none of these has proven to have modifying effects on the disease time course or progression [82]. This can be traced back to the heterogeneous nature of the disease and the need for a thorough understanding of the various disease phenotypes [83].

Despite its high spatial resolution, no MRI-based marker has been established as a “gold standard” for PD diagnosis to date. A growing mass of imaging studies, both structural and functional, over recent years have contributed to further understanding of the ongoing degenerative and compensatory processes co-occurring in the brain of these individuals. However, only limited knowledge regarding the natural history of PD and the underlying processes leading to the pathology spread can be obtained from cross-sectional imaging studies enrolling PD patients [34,80]. A summary of key findings from these studies is depicted in Figure 1. As shown in the figure, *GBA1*- and *LRRK2*-NMCs believed to be at risk demonstrate striatal DAT increases compared to NMNCs. Moreover, higher striatal within-network FC in *LRRK2*-NMCs is also observed. Upon disease manifestation, distinct patterns of brain changes are observed where both genetic PD groups show higher DAT binding compared to sPD. Functionally, *GBA1*-PDs manifest with lower striatal within-network connectivity and cortico-cortical FC gain. *LRRK2*-PDs, on the other hand, show higher striatal within-network FC and reduced FC between striatal and higher cortical regions. Additionally, *LRRK2*-PDs show reduced putaminal volumes as well as higher precentral and cerebellar GM volumes. Nevertheless, major limitations of these studies may be related to several factors such as the heterogeneity of clinical symptoms of the enrolled study samples, pooling together of different mutation carriers into one group, small sample sizes, and the effects of administered treatments, rendering the generalizability of the obtained findings from such study designs limited [38,84,85]. Additionally, it is worthwhile noting that the penetrance rates of PD-related mutations is low, hence only a minority of NMCs will phenoconvert, thus imposing additional limitations on prodromal imaging studies.

Nevertheless, the accuracy of various MRI methods was examined in detecting PD-specific changes related to disease over recent years [84,85]. Advanced imaging techniques attempted at high 3 Tesla or ultra-high (>3 Tesla) field strengths in PD such as DTI, neuromelanin-sensitive (NM) MRI, or quantitative susceptibility mapping (QSM) demonstrated promising results in detecting region-specific dopmaninergic-related abnormalities within the SN and/or nigrostriatal pathway and outside the nigrostriatal system [86,87,88,89,90,91].

The detection of genetic mutations such as *PINK*, *GBA1*, and *LRRK2* that are associated with increased risk for developing PD paved the way for a paradigm shift in PD diagnosis and the prodromal phase prior to the appearance of clinical symptoms [34,92]; providing a window period for establishing early markers with predictive validity for PD [82,92]. Several imaging studies have utilized the array of available imaging techniques to investigate candidate early imaging markers in the different prodromal subtypes, revealing distinct underlying candidate pathophysiological mechanisms [26,79,93,94]. However, more extensive research is needed to further establish these as specific early markers for disease risk prediction.

In addition to these abovementioned genetic forms of PD, mutations in the alpha-synuclein (α-syn) gene (*SNCA*) can also offer the opportunity to assess candidate neural markers in these individuals. However, its prevalence is relatively rare and nine families with *SNCA* have been identified worldwide so far [95]. Hence, the majority of imaging evidence available to date is based on a handful of case studies, rendering the generalizability of these studies somewhat limited. Pooling together imaging data from these individuals would allow for a systematic study of these individuals. The acknowledgment of PD subtypes can be considered a crucial step for characterizing candidate pathophysiological processes during the prodromal phase [94]. Pooling together of data acquired using multimodal imaging approaches including quantitative measures obtained from PET, SPECT, and MRI in combination with behavioral and clinical scores can generate multidimensional datasets, on which advanced hypothesis-free data-driven statistical techniques such as machine learning algorithms can be implemented to investigate complex association patterns and otherwise unanticipated inter-individual variations [83]. These approaches can facilitate the exploration and understanding of the biological basis of the symptoms both in the prodromal phases as well as after disease onset, possibly enabling the identification of novel endophenotypes. For example, clustering analysis was applied on multimodal imaging dataset including T1-weighted anatomical images and DTI obtained from N = 62 PD patients and N = 33 healthy controls. The used clustering approach identified three PD subtypes which differed in regional GM volumes and WM integrity parameters. Although these identified three PD groups showed comparable disease duration and severity, group 1 demonstrated bilateral tempo-parieto-occipital GM and DGM loss, as well as a widespread reduction in FA. Group 2 was mainly characterized by frontal and temporal GM reduction, while group 3 showed no detectable GM and WM changes. Based on this, the authors argued that multimodal MRI in combination with unsupervised machine learning methods is able to classify PD groups according to structural and WM features which in turn are associated with different cognitive profiles [96]. To that end, the different combination possibilities between the available conventional and advanced imaging modalities can yield valuable insights into the biological mechanisms and pathophysiology including dopaminergic dysfunction, neuroinflammation, metabolic abnormalities, and protein aggregation. This in turn can assist in quantifying risk and varied inter-individual disease progression trajectories.

Future research utilizing advanced imaging methods together with longitudinal study designs over longer follow-up periods can capture the ongoing pathological and compensatory processes, paving the way for the development of highly sensitive mechanism-oriented diagnostic markers, as well as targeted interventions and therapeutic options that can be tailored to the specific subtypes’ symptom profiles.

## Figures and Tables

**Figure 1 brainsci-13-01212-f001:**
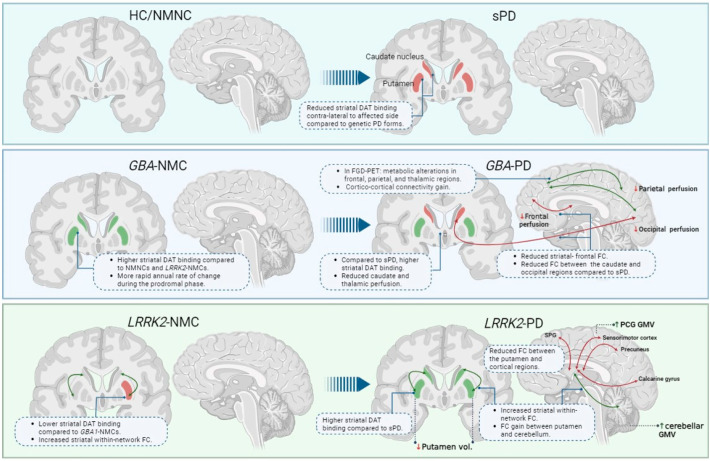
Graphical summary of significant imaging findings from nuclear imaging and MRI studies summarized in Section 1 and Section 2. Abbreviations: HC = healthy control, PD = Parkinson’s disease, NMNC = non-manifesting non-carrier, sPD = sporadic PD, NMC = non-manifesting carrier, DAT = dopamine transporter, FC = functional connectivity, vol = volume, GMV = gray matter volume.

## Data Availability

Data supporting this review content can be made available upon reasonable request.

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
