# Peer review of "Imaging Markers in Genetic Forms of Parkinson’s Disease"

_brainsci, 2023, doi:10.3390/brainsci13081212_

Round 1

Reviewer 1 Report

The manuscript by Droby et. al. reviewed brain imaging such as PET, SPECT and MRI in some genetic Parkinson’s disease. This paper focuses on GBA1 and LRRK2. The manuscript is well structured.

Major comments

  1. In hereditary Alzheimer's disease, DIAN studies have shown that amyloid-β begins to accumulate 25 years before onset. Since human alpha-synuclein PET in Parkinson’s disease was not available in the clinical situations, the authors may argue that there are no results like the DIAN study.
  2. Regarding MRI, the authors described volumetry, diffusion tensor imaging, and functional MRI. Is there any knowledge for genetic PD about conventional MRI, melatonin or nigrosome-1 images? The authors described the melatonin image in the discussion, so it would be nice to have a description in the results for genetic PD.
  3. Figure 1 can be used for education. The authors should write full spells of abbreviations such as NMC in figure legends. Figure 1 did not show glucose metabolism by [F-18]FDG PET.
  4. In Line 275, the authors wrote “Reduced striatal 18F-FDOPA with equal DAT SBR.” However, [F-18]FDOPA PET shows a change from DOPA to dopamine, which is different from dopamine transporter density.

Minor comments

  1. Line 107; “123” of 123I-FP-CIT should be superscript.
  2. Line 140; fluorodopa (FDG) –> fluorodeoxyglucose (FDG) 
  3. Line 263; CFT –> 2β-carbomethoxy-3β-(4-fluorophenyl) tropane (CFT)
  4. Line 275; FDOPA –> fluorodopa (FDOPA)
  5. Line 282; Parkinson's Progressive Marker Initiative (PPMI) –> PPMI

Author Response

Reviewer 1

The manuscript by Droby et. al. reviewed brain imaging such as PET, SPECT and MRI in some genetic Parkinson’s disease. This paper focuses on GBA1 and LRRK2. The manuscript is well structured.

Response: We wish to thank the esteemed reviewer for reviewing our manuscript, and for the valuable comments and suggestions.

Major comments

  1. In hereditary Alzheimer's disease, DIAN studies have shown that amyloid-β begins to accumulate 25 years before onset. Since human alpha-synuclein PET in Parkinson’s disease was not available in the clinical situations, the authors may argue that there are no results like the DIAN study.

Response: We thank the reviewer for this comment. We have inserted the following statement to the introduction to further highlight the need for highly sensitive and specific imaging markers for early detection of PD subtypes.

Unlike Alzheimer's disease (AD), where imaging markers for tracing amyloid-β up to 25 years prior to clinical symptoms onset are available, to date,  no synuclein tracer has been validated thus other imaging modalities are used in PD.”

  1. Regarding MRI, the authors described volumetry, diffusion tensor imaging, and functional MRI. Is there any knowledge for genetic PD about conventional MRI, melatonin or nigrosome-1 images? The authors described the melatonin image in the discussion, so it would be nice to have a description in the results for genetic PD.

Response: As suggested by the reviewer, we have added a summary of neuromelanin MRI studies both in genetic forms of PD and in carriers of PD-related genetic mutations to the corresponding sub-sections of the body text.

  1. Figure 1 can be used for education. The authors should write full spells of abbreviations such as NMC in figure legends. Figure 1 did not show glucose metabolism by [F-18]FDG PET.

Response: We appreciate this valuable comment. We have further modified the Figure as suggested highlighting FDG-PET findings as well. Moreover, we have defined all abbreviations in the figure legend.

  1. In Line 275, the authors wrote “Reduced striatal 18F-FDOPA with equal DAT SBR.” However, [F-18]FDOPA PET shows a change from DOPA to dopamine, which is different from dopamine transporter density.

Response: We thank the reviewer for pointing this point to our attention. This was not conveyed correctly. In fact, Wile et al reported preserved F-DOPA binding and reduced DAT SBRs in LRRK-NMCs compared with healthy controls. We have modified this sentence and added further explanation of these results for further clarification. This part now reads as follows:

“Comparable striatal 18F-flurodopa (FDOPA) binding and reduced DAT SBRs were observed in N=25 LRRK2-NMCs compared to N=35 healthy controls; however, these observed levels in the NMCs groups were significantly higher than those of LRRK2-PD [48] The authors of this study speculated that this finding might be representative of a primary effect on DAT function; where preserved FDOPA can persist up to symptoms onset [48].”

  1. Minor comments

- Line 107; “123” of 123I-FP-CIT should be superscript.

- Line 140; fluorodopa (FDG) –> fluorodeoxyglucose (FDG) 

- Line 263; CFT –> 2β-carbomethoxy-3β-(4-fluorophenyl) tropane (CFT)

- Line 275; FDOPA –> fluorodopa (FDOPA)

- Line 282; Parkinson's Progressive Marker Initiative (PPMI) –> PPMI

Response: We thank the reviewer for pointing out to these mistakes. We have corrected those accordingly.

Reviewer 2 Report

Review of a manuscript “Imaging markers in genetic Parkinson's disease” by Amgad Droby and coauthors submitted to “Brain Sciences”

Parkinson’s disease is a sever neurodegenerative disease, second after Alzheimer’s disease. There are no efficient medications affecting the course of this disorder, and the source of reliable markers for early identification of the early step of this illness.  Finding of biomarkers for Parkinson’s disease is an urgent need since it may help to begin specific treatment at the early stage of the disorder. The authors of the review discuss the application of imaging markers in the genetic forms of Parkinson’s disease. This is an important biomedical issue, and the manuscript will be useful of the readership of the journal “Brain Sciences”. The following corrections and additions should be made.

Title and Abstract:

The authors may consider changing the title “Imaging markers in genetic Parkinson's disease” to “Imaging markers in genetic forms of Parkinson's disease” and in the Abstract and other sites of the text replace “neuro degenerative disorder” on “neurodegenerative disorder”.

Background:

Line 25: “..common neurodegenerative disorder [1.” Should be corrected as “common neurodegenerative disorder [1].

Line 28: “…reaching 1% in individuals above 60 years of age [2] (Nussbaum & Ellis, 2003).” Pease be consistent in using references format :” reaching 1% in individuals above 60 years of age [2].

1.       Imaging findings in genetic Parkinson’s disease patients

“In a retrospective analysis, Lee et al. modeled the temporal trajectories of measured…” “…unlike the reported findings by McNeill et al.,” and in other places of the text:” Please, be consistent in using references format.

Discussion

Lines 391-392:”This can be traced back to the heterogeneous nature of the disease and the need for a thorough understanding of the various disease phenotypes.” Fter this sentence the authors should add the reference to a recent PD review: ”Biomarkers in Parkinson’s Disease”. Chapter in a book Peplow P.V.,et al., eds. “Neurodegenerative Diseases Biomarkers. 2022. Neuromethods, vol 173. pp 155-180. Humana, NY.c https://link.springer.com/protocol/10.1007/978-1-0716-1712-0_7

Lines 397-398 :”However, only limited knowledge regarding the natural history of PD and the underlying processes leading to the pathology spread can be obtained from cross- sectional imaging study enrolling PD patients.” The authors should add references after this sentence.

Lines 416-420:

The authors repeat the full names and abbreviated forms for the words that have been abbreviated earlier, for example:

Substantia nigra- on line 30, DTI – on line 180, etc. One abbreviation is enough, after which the term can be used in abbreviated form through the whole text.

Lines 423-425: “The detection of several genetic mutations associated with the risk for developing PD paved the way for a paradigm shift in PD diagnosis, and the prodromal phase prior to the appearance of clinical symptoms [34, 84], providing a window period for establishing early markers with predictive validity for PD [74, 84].” The authors should be more specific, explaining what they mean by “ several genetic mutations”

At any appropriate site of Discussion the authors should add more information about the status of SNCA as an imaging marker for PD.  

Overall, this is an interesting review which will be helpful for scholars working in Parkinson’s disease area.

Author Response

Comments and Suggestions for Authors

Reviewer 2

Review of a manuscript “Imaging markers in genetic Parkinson's disease” by Amgad Droby and coauthors submitted to “Brain Sciences”

Parkinson’s disease is a sever neurodegenerative disease, second after Alzheimer’s disease. There are no efficient medications affecting the course of this disorder, and the source of reliable markers for early identification of the early step of this illness.  Finding of biomarkers for Parkinson’s disease is an urgent need since it may help to begin specific treatment at the early stage of the disorder. The authors of the review discuss the application of imaging markers in the genetic forms of Parkinson’s disease. This is an important biomedical issue, and the manuscript will be useful of the readership of the journal “Brain Sciences”. The following corrections and additions should be made.

Response: We would like to thank the esteemed reviewer for the positive assessment of our manuscript and for the constructive feedback and suggestions.

Title and Abstract:

  1. The authors may consider changing the title “Imaging markers in genetic Parkinson's disease” to “Imaging markers in genetic forms of Parkinson's disease” and in the Abstract and other sites of the text replace “neuro degenerative disorder” on “neurodegenerative disorder”.

Response: We thank the reviewer for the suggestion. We have modified the tittle as suggested. We also made the correction the phrase “neurodegenerative”.

Background:

  1. Line 25: “..common neurodegenerative disorder [1.” Should be corrected as “common neurodegenerative disorder [1].

Response: We have corrected the reference style.

  1. Line 28: “…reaching 1% in individuals above 60 years of age [2] (Nussbaum & Ellis, 2003).” Pease be consistent in using references format :” reaching 1% in individuals above 60 years of age [2].

Response: We thank the reviewer for pointing this out. We have corrected the reference style.

  1. Imaging findings in genetic Parkinson’s disease patients

“In a retrospective analysis, Lee et al. modeled the temporal trajectories of measured…” “…unlike the reported findings by McNeill et al.,” and in other places of the text:” Please, be consistent in using references format.

Response: We agree with the reviewer regarding this comment. We have now adjusted the reference format and made it consistent throughout, we now think that this would makes it easier for the reader to look up the cited paper and easily locate in the reference list.

Discussion

  1. Lines 391-392:”This can be traced back to the heterogeneous nature of the disease and the need for a thorough understanding of the various disease phenotypes.” Fter this sentence the authors should add the reference to a recent PD review: ”Biomarkers in Parkinson’s Disease”. Chapter in a book Peplow P.V.,et al., eds. “Neurodegenerative Diseases Biomarkers. 2022. Neuromethods, vol 173. pp 155-180. Humana, NY.c https://link.springer.com/protocol/10.1007/978-1-0716-1712-0_7

Response: We thank the reviewer for the suggestion. We have added the references.

  1. Lines 397-398 :”However, only limited knowledge regarding the natural history of PD and the underlying processes leading to the pathology spread can be obtained from cross- sectional imaging study enrolling PD patients.” The authors should add references after this sentence.

Response: We thank the reviewer for the suggestion. We have inserted references for this statement in the discussion.

  1. Lines 416-420:

The authors repeat the full names and abbreviated forms for the words that have been abbreviated earlier, for example:

Substantia nigra- on line 30, DTI – on line 180, etc. One abbreviation is enough, after which the term can be used in abbreviated form through the whole text.

Response: We thank the reviewer for pointing this out. We have omitted the full names of previously abbreviated phrases.

  1. Lines 423-425: “The detection of several genetic mutations associated with the risk for developing PD paved the way for a paradigm shift in PD diagnosis, and the prodromal phase prior to the appearance of clinical symptoms [34, 84], providing a window period for establishing early markers with predictive validity for PD [74, 84].” The authors should be more specific, explaining what they mean by “several genetic mutations”

Response: We have modified this sentence further specifying which genetic mutations was referred to. This part of the discussion now reads as follows: “The detection of genetic mutations such as PINK, GBA1, and LRRK2 that associated with increased risk for developing PD paved the way for a paradigm shift in PD diagnosis, and the prodromal phase prior to the appearance of clinical symptoms [34, 84]”.

  1. At any appropriate site of Discussion the authors should add more information about the status of SNCA as an imaging marker for PD.  

Response: We agree with the reviewer that SNCA requires to be addressed in the discussion. Based on the suggestion we have inserted the following paragraph to the discussion.

“In addition to these abovementioned genetic forms of PD, mutations in the alpha-synuclein (α-syn) gene (SNCA) can also offer the opportunity to assess candidate neural markers in these individuals. However, its prevalence is relatively rare where nine families with SNCA have been identified world-wide so far [93]. Hence, the majority of imaging evidence available to date is based on a handful of case studies rendering the generalizability of these studies somewhat limited. Pooling together imaging data from these individuals would allow for a systematic study of these individuals.

Round 2

Reviewer 1 Report

The manuscript by Droby et. al. has been well revised, and I think that the paper is acceptable for publication.

Author Response

We thank the reviewer for the efforts and positive feedback on our manuscript.